# N-Terminal Pro-Brain Natriuretic Peptide and Right Ventricular Diameter Are Related to Aspirin Resistance in Coronary Artery Disease Patients

**DOI:** 10.3390/medicina57070706

**Published:** 2021-07-12

**Authors:** Kamila Marika Cygulska, Łukasz Figiel, Dariusz Sławek, Małgorzata Wraga, Marek Dąbrowa, Jarosław D. Kasprzak

**Affiliations:** 1Department of Cardiology, Medical University of Lodz, Bieganski Hospital, 1/5 Kniaziewicza, 91-347 Lodz, Poland; medluk@wp.pl (Ł.F.); szpital@bieganski.com.pl (D.S.); mwraga@op.pl (M.W.); kasprzak@ptkardio.pl (J.D.K.); 2Department of Biopharmacy, Chair of Biopharmacy, Medical University of Lodz, 1/5 Kniaziewicza, 91-347 Lodz, Poland; marek.dabrowa@aflofarm.pl

**Keywords:** acetylsalicylic acid, coronary disease, N-terminal pro-brain natriuretic peptide, aspirin resistance

## Abstract

*Background and Objectives:* Resistance to ASA (ASAres) is a multifactorial phenomenon defined as insufficient reduction of platelet reactivity through incomplete inhibition of thromboxane A2 synthesis. The aim is to reassess the prevalence and predictors of ASAres in a contemporary cohort of coronary artery disease (CAD) patients (pts) on stable therapy with ASA, 75 mg o.d. *Materials and Methods:* We studied 205 patients with stable CAD treated with daily dose of 75 mg ASA for a minimum of one month. ASAres was defined as ARU (aspirin reaction units) ≥550 using the point-of-care VerifyNow Aspirin test. *Results:* ASAres was detected in 11.7% of patients. Modest but significant correlations were detected between ARU and concentration of N-terminal pro-brain natriuretic peptide (NT-proBNP) (*r* = 0.144; *p* = 0.04), body weight, body mass index, red blood cell distribution width, left ventricular mass, and septal end-systolic thickness, with trends for left ventricular mass index and prothrombin time. In multivariate regression analysis, log(NT-proBNP) was identified as the only independent predictor of ARU—partial *r* = 0.15, *p* = 0.03. Median concentrations of NT-proBNP were significantly higher in ASAres patients (median value 311.4 vs. 646.3 pg/mL; *p* = 0.046) and right ventricular diameter was larger, whereas mean corpuscular hemoglobin concentration was lower as compared to patients with adequate response to ASA. *Conclusions:* ASAres has significant prevalence in this contemporary CAD cohort and NT-proBNP has been identified as the independent correlate of on-treatment ARU, representing a predictor for ASAres, along with right ventricular enlargement and lower hemoglobin concentration in erythrocytes.

## 1. Introduction

Modulation of thrombocytes’ reactivity with antiplatelet drugs has become a cornerstone of modern cardiovascular pharmacotherapy. Conversion of arachidonic acid to thromboxane A2 (TXA2) by cyclooxygenase-1 (COX-1) enzyme contributes to proper platelet aggregation. This enzyme catalyzes the synthesis of prostaglandin H2 from arachidonic acid, further converted to TXA2 by TXA2 synthase. Aspirin (ASA), a commonly used antiplatelet, irreversibly blocks COX-1 by acetylation of serine at position 529 of COX-1, modifying the catalytic center of the enzyme and preventing binding of arachidonic acid. However, ASA does not produce a sufficient antiplatelet effect in all patients. In the case of resistance to ASA (ASAres), TXA2 synthesis is not sufficiently inhibited and residual risk of thrombus formation in coronary arteries remains [1,2]. COX-1 is an active enzyme in many tissues, including kidneys, platelets and stomach. Multiple factors may contribute to ASAres: incorrect dose of ASA, drug interactions, changes in COX-1 gene and other genes connected with thromboxane synthesis, increased non-platelet thromboxane production and excessive platelet activation [3,4]. Currently, ASAres can be assessed at the bedside with point-of-care devices, e.g., the VerifyNow analyzer. In the VerifyNow Aspirin test arachidonic acid is used as the agonist to express residual activity of blocked platelets in Aspirin Reaction Units (ARU). Previously reported risk factors for ASAres are multifactorial, including clinical correlates such as: age, type 2 diabetes mellitus (DM2) [5,6], elevated creatinine [7,8] and glucose values, dyslipidemia [9], body mass index (BMI) [10] and smoking [11].

We decided to analyze the correlation of ASAres with echocardiographic parameters due to the previously described relationship of ASAres with body weight and multiple risk factors for atherosclerotic heart disease. Previous publications have confirmed the relationship between body weight, BMI and left ventricular mass index (LVMI) [12].

The aim of our study was to assess the prevalence of ASAres in patients with coronary artery disease (CAD) chronically treated with standard ASA dose, to study the correlation of on-treatment ARU with clinical and echocardiographic variables and to identify potential predictors of ASAres.

## 2. Materials and Methods

### 2.1. Study Group

This prospective single-center study recruited 205 patients (ASAsens—aspirin-sensitive patients, and ASAres—aspirin-resistant patients) hospitalized in our department from October 2018 to February 2019 with diagnosed stable CAD. The patients’ flow diagram is shown in Figure 1.

All patients were treated with a stable regimen of 75 mg ASA o.d. for a minimum of one month following current guidelines—directed coronary artery disease medical therapy [13]. Exclusion criteria included: recent (up to 2 months) acute coronary syndrome, cancer, dermatological disease, epilepsy or other chronic neurological diseases, exacerbation of allergic disease, rheumatoid arthritis, alcoholism, drug addiction, vegetarianism, veganism and other specific diets, and known thrombophilia. The study was approved by the local ethics committee at the Medical University of Lodz in Poland (Number RNN/415/18/KE). All patients provided written consent to participate in our study.

### 2.2. Platelet Function Measurement

We measured ASAres using VerifyNow (Werfen, Barcelona, Spain) point-of-care platelet function analyzer [14,15]. The incubation time after blood donation was 30 min. Platelet function on treatment was quantified using VerifyNow Aspirin tests containing arachidonic acid as agonist. In this method microparticles are coated with fibrinogen aggregate in proportion to the number of GP IIa/IIIb receptors of activated platelets. Platelet aggregation is reduced with adequate antiplatelet action of aspirin and platelet reactivity is reported in Aspirin Reaction Units (ARU) ranging from 350 to 700 ARU. The values of ARU lower than 550 indicate effective aspirin therapy, while on-treatment ARU greater than or equal to 550 ARU confirms aspirin resistance.

### 2.3. Clinical Data

We compared atherosclerotic risk factors (overweight, smoking, a hypertension, dyslipidemia, hypercholesterolemia, diabetes), the history of treatment and comorbidity profiles in both groups. Blood tests including lipid panel, creatinine, glucose, hemoglobin A1c (HbA1c), N-terminal pro-brain natriuretic peptide (NT-proBNP), C-reactive protein (CRP), coagulogram and morphology were performed on an empty stomach using the standard method in appropriate test tubes.

### 2.4. Transthoracic Echocardiography

The echocardiographic studies were performed by the same team of the echocardiographic laboratory in I Department of Cardiology Medical University of Lodz according to guidelines of the Echocardiography Section of the Polish Cardiac Society, and we measured standard diameters of cardiac cavities, left ventricular end-diastolic diameter (LVEDD), posterior wall end-diastolic thickness (PWTd) and interventricular septal end-diastolic thickness (IVSTd); left ventricular mass (LVM) and (LVMI) were calculated [16]. Due to the assessment of the laboratory relationship of the heart failure index (NT-proBNP) with ASAres, we decided to assess the relationship of ASAres with end-diastolic diameter of the right ventricle. Proximal right ventricular outflow tract was measured in parasternal long-axis view as a marker of RV enlargement.

### 2.5. Statistical Analysis

We analyzed the data using MedCalc Statistical Software version 12.2.1.0 (MedCalc Software bvba, Ostend, Belgium; 2012). We used the D’Agostino–Pearson’s test to confirm the normal distribution of data. We used the independent samples t-test or Mann–Whitney test as non-parametric equivalent to confirm the differences between the subgroups of ASAres and ASAsens patients. The chi-square test was used to test the statistical significance of associations between categorical variables. We used the normal approximation (Lentner, 1982) to calculate the *p*-value, with a *p*-value less than 0.05 accepted as significant. We analyzed the relationships between ARU and the independent clinical variables (usually presenting non-normal distribution) using Spearman’s coefficient of rank correlation. Multivariate regression analysis was applied to determine the relationship between ASAres and independent risk factors.

## 3. Results

Median values for ARU in the study group were 427 (range 327–660) and the prevalence of ASAres was 11.7% (24/205 pts). We included 36.6% females, average age 68.2 (9.7) years, average BMI 27.3 (4.6) kg/m^2^. The study group characteristics regarding the entire study cohort and the subgroups defined by the presence or absence of ASAres are presented in Table 1, and echocardiographic parameters are shown in Table 2.

We identified several novel correlates of ASAres which can contribute to better understanding of the significant phenomenon of aspirin resistance. On-treatment ARU values showed multiple correlations with studied clinical data as shown in Table 3, with positive correlations between ARU and NT-proBNP, body weight, body mass index, red blood cell distribution width, left ventricular mass, and interventricular septal end-systolic thickness with trends for left ventricular mass index, C-reactive protein and prothrombin time.

In multivariate regression analysis excluding echocardiographic data (*n* = 205), the only independent predictor of ARU was log(NT-proBNP)—partial *r* = 0.15, *p* = 0.03. With echocardiographic data included in the multivariate analysis (*n* = 186), the optimal model (*p* < 0.001, multiple *r* = 0.35) was composed of three independent predictors for ARU: log(NT-proBNP), body mass and interventricular septal end-systolic thickness.

We subsequently attempted to identify predictors of ASAres status defined as a dichotomous variable. Significant differences between ASAres/ASAsens subjects were demonstrated for several factors: NT-proBNP, mean corpuscular hemoglobin concentration, and right ventricular diastolic diameter, and a few others showed borderline trends: red blood cell distribution width, left ventricular ejection fraction, interventricular septal end-systolic thickness, left atrial dimension and left ventricular mass index. Other variables including diabetes and medical therapy profile, e.g., proton pump inhibitors, were similarly distributed in ASA responders and non-responders.

Mann–Whitney’s test showed that median NT-proBNP concentrations were significantly higher (median 311.4 vs. 646.3 pg/mL; *p* = 0.046) in ASAres patients. The ROC curve for log(NT-proBNP) presents the area under the curve AUC = 0.625 (95% CI: 0.555 to 0.692; *p* = 0.03) with a threshold value of 2.51 providing sensitivity of 75.0% and specificity 51.4%, negative predictive value of 94.0% and positive predictive value of 17.0% for ASAres. VerifyNow tests are used as screening tests for patients with ASAres. Our study confirms that non-elevated NT-proBNP is correlated with low likelihood of ASAres.

Multiple logistic regression failed to improve significantly the prediction of ARU due to strong correlations between RV, NT-proBNP and (negative) MCHC. Figure 2 presents ROC curves for log (NT-proBNP).

## 4. Discussion

The main finding of our study is identification of ASAres rate in the contemporary population of Polish CAD patients and possible association of ARU measured on treatment with 75 mg ASA per day with natriuretic peptides and RVD. ASAres has been identified in the population with stable CAD [17] and is observed more often among people with acute coronary syndromes (ACS) [18]. In the Polish population, ASAres was observed in more than 1 patient in 10 patients in the literature [19]. In our study, we did not confirm the differences between groups ASAres and ASAsens depending on the risk factors mentioned in the literature: age, type 2 diabetes mellitus, elevated creatinine, glucose value, dyslipidemia, body mass index (BMI) and smoking. The differences may be due to various reasons, including study group characteristics, selection of variables included in the analysis, and background medication but also potentially laboratory methods. The differences in creatinine results were due to patient selection; in our study there were no significant differences in stage of renal failure [7]. We did not show an association with aspirin resistance and glucose levels, probably because few with diabetes patients had been included in our study [10]. We did not confirm a connection with aspirin resistance and age or smoking, as in the study by Xian-Feng Liu et al. [6]. We confirm a positive correlation between aspirin resistance and BMI found in the study of Ertugrul et al. [10]. We showed positive correlations between ARU and body mass index. All risk factors mentioned in the introduction were shown in Table 1. We showed positive correlations between ARU and both body weight and body mass index. In our discussion we will try to show probable mechanisms of association of ASAres with heart failure, obesity, left ventricular hypertrophy and right ventricular diameter.

### 4.1. Heart Failure and Left Ventricular Hypertrophy in Patients with Aspirin Resistance

NT-proBNP value is associated with heart failure. Previous research showed that the incidence of aspirin resistance increased in patients with chronic heart failure (CHF) and decompensated type 2 diabetes mellitus (DM2). PLUTO-CHF trial observed higher platelet activity in patients with CHF treated with ASA [20,21]. However, there are results presented in the literature that find that the highest platelet aggregation is found in functional NYHA III class patients. [22] The potential causes of ASAres in CHF and DM are: lower endothelial nitric oxide synthesis, higher platelet activity, changed platelets by cholesterol or increase in intra-platelet calcium level [23].

The process of heart failure is connected with the renin–angiotensin–aldosterone system (RAAS) activation, which contributes to increased synthesis of angiotensin II. We can assume that activation of TXA2 is stimulated by angiotensin II. Thromboxane A2 promotes platelet aggregation through the thromboxane prostanoid (TP) receptor and this overstimulation may be linked to aspirin resistance [24]. The scheme of the potential aspirin resistance mechanism in patients with heart failure is shown in Figure 3.

Some studies have shown the decrease of platelet aggregation by angiotensin converting enzyme (ACE) inhibitors. The inhibition of TXA2 synthesis was demonstrated for captopril and fosinopril [25,26,27]. Similar antiaggregation effects were shown for angiotensin receptor blockers, e.g., irbesartan had a similar effect to fosinopril [28].

We found a correlation between on-treatment ARU value and LVM which can, therefore, be hypothetically linked to increased TXA2 activity in ASAres patients—it was demonstrated that the TP receptor mediates angiotensin II-dependent hypertension and left ventricular hypertrophy in mouse model [29].

### 4.2. The Impact of Aspirin on the Right Ventricle

Our research showed that proximal right ventricular outflow tract diameter was correlated with aspirin resistance. Increased isoprostane (8-isoPGF2α or 8-isoF), an oxidative stress marker, is connected with right ventricular dilatation and heart failure. One of the hypotheses regarding aspirin resistance is based on the non-enzymatic effect of isoprostane on thromboxane A2 receptors on the platelet surface. Despite inhibition of TXA2, platelets activated by TP receptor and platelet aggregation continue despite treatment with ASA. It may be suggested that increased oxidative stress can lead to sustained platelet activity independent of the blocked COX-1 pathway [30]. Beside putative biochemical links, larger RV diameter is common in patients with larger body size and mass index. We identified a positive correlation of residual platelet activity and body mass index. Overweight patients have increased platelet activity associated with the COX-1 enzyme and increasing the dose of ASA to 325 mg reduced the risk of ASAres in obese patients [31]. Obesity is associated with proinflammatory cytokines and increased COX-2 enzyme activity, resulting in increased prostaglandin PGE2, which promotes platelet activity and the risk of thrombosis by the prostaglandin E3 (EP3) receptor (Figure 4) [32,33].

### 4.3. Aspirin Resistance vs. Pseudoresistance in Everyday Practice

During aspirin treatment, special attention should be paid to patients with heart failure and obesity—we did not study these but they are common. We can assume that pill formulations may influence bioavailability of ASA. Evidence exist that aspirin with enteric coating may limit the efficacy of aspirin. [34] Other approaches to reduce gastrointestinal side effects can be used, e.g., aspirin formulation based on lipids with modified release (aspirin PL2200) or single pill ASA-glycine combination [35]. Searching for risk factors of aspirin resistance is important as this phenomenon is linked with risk of recurring cardiovascular events in CAD, even though routine platelet function tests are not recommended by therapeutic guidelines [36]. Our study may contribute to considering platelet function tests in patients with ASA treatment failure, heart failure and obesity, in order to improve treatment effectiveness.

### 4.4. Study Limitations

Study design was a single center and subject count was moderate. Proximal RVOT was chosen as the only RV size marker because of its common use in community echocardiographic laboratories in Poland. Platelet function is a complex activity and alternative tests can be used for assessing reactivity—for practical reasons we decided to use only one simple yet clinically applicable point-of-care-specific test. Close to 30% of our patients received dual antiplatelet therapy which could represent an additional confounding factor. We did not analyze specific ASA formulation but enteric-coated pills represent the vast majority of formulation used in secondary prevention in Poland.

## 5. Conclusions

The prevalence of ASAres in a contemporary cohort of patients with coronary disease treated with 75 mg daily ASA dose was 11.7%. We identified NT-proBNP as the only independent correlate of on-treatment ARU, and a predictor for ASAres, along with right ventricular enlargement and lower hemoglobin concentration in erythrocytes. The relationship of thrombocytes’ response to ASA in patients with heart failure warrants further studies in the clinical setting.

## Figures and Tables

**Figure 1 medicina-57-00706-f001:**
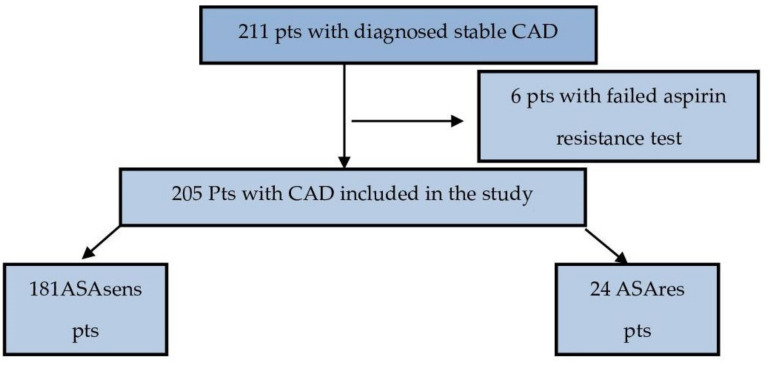
Patient inclusion flow diagram. ASAsens pts—aspirin sensitive patients, ASAres. pts—aspirin-resistant patients.

**Figure 2 medicina-57-00706-f002:**
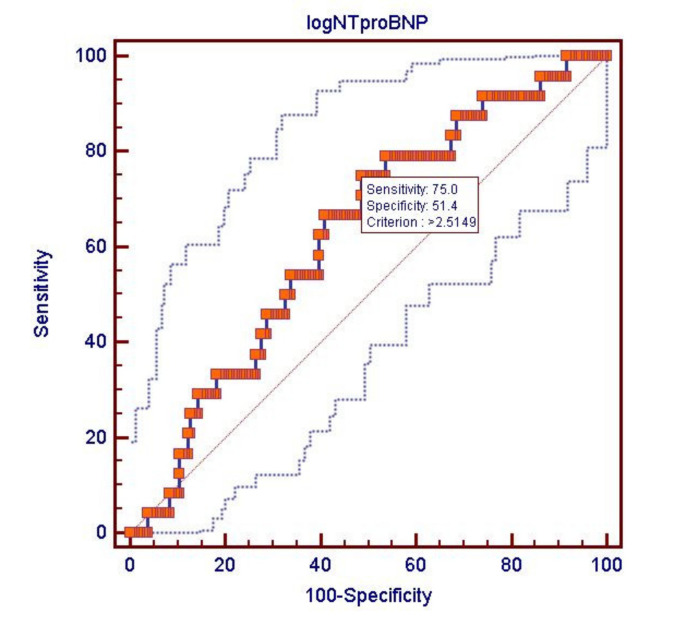
Receiver-operating characteristic curve for log (NT-proBNP) as a predictor of resistance to acetylsalicylic acid.

**Figure 3 medicina-57-00706-f003:**
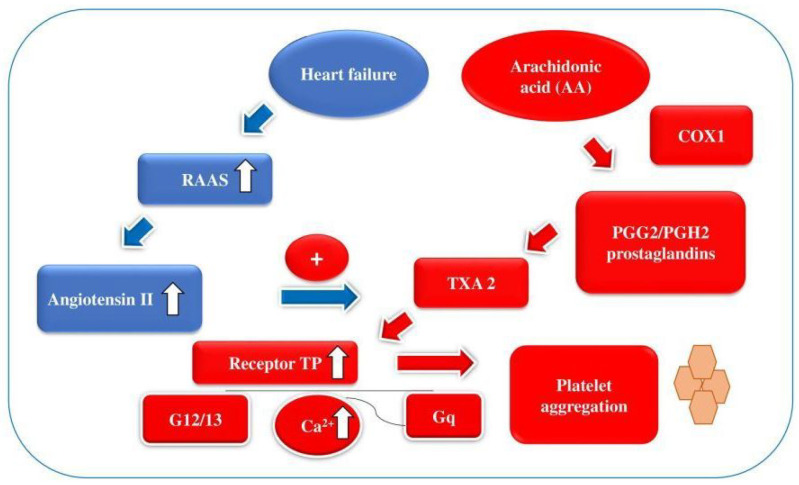
Potential mechanisms of aspirin resistance in patients with heart failure. COX-1—cyclooxygenase 1, G protein families—G12/13, Gq, Ca ^2+^—calcium ion, HF—heart failure, PGG2—prostaglandin G2, PGH2—prostaglandin H2, RAAS—renin–angiotensin–aldosterone system, TP receptor—thromboxane prostanoid receptor, TXA2—thromboxane A2.

**Figure 4 medicina-57-00706-f004:**
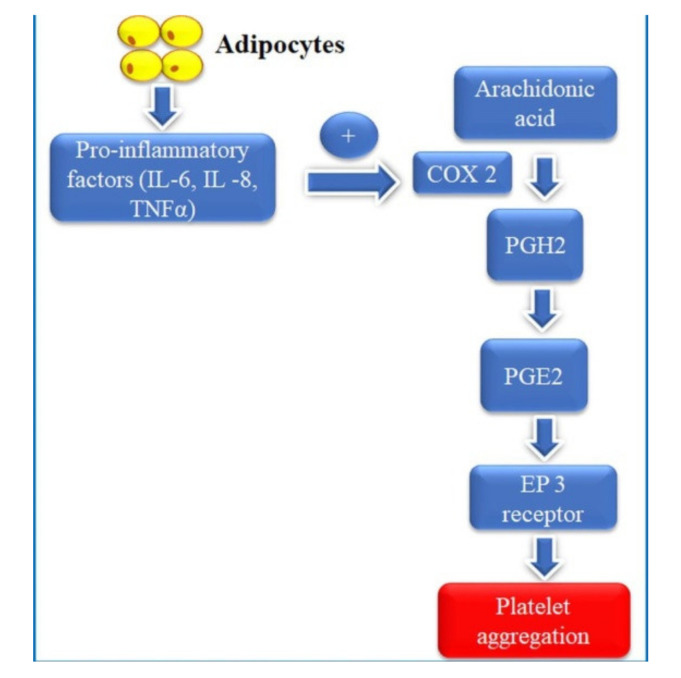
Potential interactions between obesity and high on-treatment platelet reactivity. IL—interleukin, COX-2—cyclooxygenase 2, PGH2—prostaglandin H2, PGE2—prostaglandin E2, EP3 receptor—prostaglandin E3 receptor, TNF—α—tumor necrosis factor alpha.

**Table 1 medicina-57-00706-t001:** The study group characteristics. *p*-value refers to comparisons between the aspirin-resistant (ASAres) positive and aspirin-sensitive (ASAsens) patient subgroups.

Parameters	Total (*n* = 205)	ASAsens (*n* = 181)	ASAres (*n* = 24)	*p*
Age	68.2 (9.7)	67.9 (9.7)	70.8 (9.1)	0.17
Sex (female) (%)	36.6	38.1	25.0	0.30
Body weight (kg)	78.2 (16.3)	77.9 (15.8)	83.8 (18.9)	0.13
Body mass index (kg/m^2^)	27.3 (4.6)	27.2 (4.43)	28.7 (5.7)	0.20
NT-proBNP (pg/mL)	351.2 (136.8–1069.8)	311.4 (127.7–1051.0)	646.3 (309.8–2389.5)	0.046
Creatinine (mg/dL)	0.97 (0.82–1.14)	0.96 (0.82–1.14)	1.03 (0.87–1.19)	0.50
Glucose (mg/dL)	99.0 (90.0; 120.75)	99.0 (90.75; 120.75)	103.5 (89.5; 130.0)	0.75
Glycated hemoglobin A1c (%)	5.8 (5.4; 6.4)	5.8 (5.4; 6.5)	5.8 (5.5; 6.2)	0.88
Total cholesterol (mg/dL)	148.5 (41.4)	149.7 (42.0)	139.1 (34.8)	0.25
LDL cholesterol (mg/dL)	65.0 (50.8–89.8)	67.0 (50.0–92.3)	63.0 (54.0–82.5)	0.58
HDL cholesterol (mg/dL)	47.9 (38.6–56.1)	48.2 (38.7–56.1)	40.9 (35.9–57.9)	0.45
Triglycerides (mg/dL)	112.0 (86.8–153.3)	115.0 (85.8–157.5)	98.5 (88.5–113.0)	0.17
C-reactive protein (mg/L)	2.0 (0.8–4.1)	2.0 (0.8–4.4)	1.9 (0.9–3.7)	0.80
White blood cells (10^3^ µL)	7.7 (6.4–9.1)	7.7 (6.4–9.1)	7.8 (6.2–9.7)	0.94
Red blood cells (10^6^ µL)	4.6 (4.2–4.9)	4.5 (4.2–4.9)	4.7 (4.4–5.0)	0.43
Hemoglobin (g/dL)	13.9 (13.0–15.0)	13.9 (12.9–14.9)	14.1 (13.2–15.2)	0.50
Hematocrit (%)	41.8 (38.9; 44.6)	41.8 (38.9; 44.4)	42.1 (39.8; 46.1)	0.28
Mean corpuscular volume (fL)	91.6 (89.0–94.7)	91.6 (89.3–94.4)	91.3 (87.5–97.5)	0.91
Mean corpuscular hemoglobin (pg)	30.7 (29.7–31.7)	30.8 (29.8–31.7)	30.2 (29.0–32.3)	0.47
MCHC (g/dL)	33.4 (33.0–33.9)	33.5 (33.0–33.9)	33.2 (32.7–33.5)	0.02
RDW (%)	14.0 (13.4–14.8)	14.0 (13.3–14.7)	14.1 (13.8–15.7)	0.06
RDW—SD (fL)	44.6 (42.4–47.7)	44.2 (42.4–47.4)	46.2 (42.9–47.7)	0.14
Platelets (10^3^ µL)	206.4 (60.6)	206.6 (60.6)	204.8 (61.8)	0.90
Mean platelet volume (fL)	9.5 (8.8; 10.1)	9.5 (8.8; 10.1)	9.5 (9.1; 10.1)	0.96
Activated partial thromboplastin time (s)	28.9 (26.8–31.7)	29.1 (26.9–31.7)	28.6 (26.3–32.1)	0.73
Prothrombin time (s)	12.1 (11.3–12.9)	12.0 (11.3–12.7)	12.4 (11.8–14.3)	0.11
Fibrinogen(mg/dL)	313.0 (270.8; 380.0)	315.0 (270.8; 378.5)	301.5 (270.0; 385.5)	0.88
Hypertension (%)	86.8	87.8	79.2	0.39
Diabetes (%)	39.5	39.8	37.5	0.99
Chronic kidney disease (%)	13.2	13.3	12.5	0.83
Severe valve disease (%)	3.9	3.3	8.3	0.53
Previous stroke (%)	10.2	10.5	8.3	0.98
Previous myocardial infarction (%)	36.6	37.0	33.3	0.90
Previous coronary angioplasty (%)	63.9	63.5	70.8	0.50
Smoking (%)	12.7	10.5	20.8	0.34
Statins (%)	93.2	93.4	91.7	0.90
Angiotensin-converting enzyme inhibitors (%)	76.1	75.1	83.3	0.53
Angiotensin II receptor blockers (%)	12.2	13.3	4.2	0.32
Beta—blockers (%)	85.9	87.3	75.0	0.19
Proton pump inhibitors (%)	65.4	64.6	70.8	0.71
Dual antiplatelet therapy (%)	28.8	27.6	37.5	0.44

MCHC—mean corpuscular hemoglobin concentration, RDW—red blood cell distribution width, SD—standard deviation. Data are shown as mean (standard deviation) or median [interquartile range] according to distribution.

**Table 2 medicina-57-00706-t002:** Echocardiographic parameters in the study group. *p*-value refers to comparisons between the aspirin-resistant (ASAres) positive and aspirin-sensitive (ASAsens) patient subgroups.

Parameters	Total (*n* = 181)	ASAsens (*n* = 165)	ASAres (*n* = 21)	*p*
LVEF (%)	52.0 (40.0–58.0)	52.0 (40.0–58.0(	47.5 (39.0–52.0)	0.06
LVEDD (mm)	50.0 (45.0–54.0)	49.0 (45.0–54.0)	52.0 (47.8–56.5)	0.12
LVESD (mm)	36.3 (9.8)	36.0 (9.9)	39.0 (9.0)	0.14
LAD (mm)	44.1 (6.5)	43.8 (6.3)	48.5 (8.0)	0.10
IVSTs (mm)	14.0 (13.0–16.0)	14.0 (13.0–15.0)	15.0 (14.0–17.0)	0.06
IVSTd (mm)	12.0 (11.0–13.0)	12.0 (11.0–13.0)	12.0 (11.0–14.0)	0.37
PWTs (mm)	14.0 (13.0–15.0)	14.0 (13.0–15.0)	14.0 (13.0–16.0)	0.60
PWTd (mm)	11.0 (11.0–12.0)	11.0 (11.0–12.0)	11.0 (10.0–13.0)	0.98
RVD (mm)	29.7 (4.4)	29.4 (4.1)	32.0 (5.9)	0.02
LVMI (g/m^2^)	138.0 (117.0–161.0)	136.0 (116.0–160.3)	147.0 (136.3–167.3)	0.10
LVM (g)	268.0 (220.0–316.0)	266.0 (220.0–315.3)	285.0 (240.5–368.3)	0.15
TAPSE (mm)	21.0 (18.0–24.0)	21.5 (18.0–23.0)	21.0 (18.0–24.0)	0.99
RWT	0.5 (0.4–0.5)	0.5 (0.4–0.5)	0.5 (0.4–0.5)	0.36

IVSTd—interventricular septal end diastolic thickness, IVSTs—interventricular septal end systolic thickness, LVEF—left ventricular ejection fraction, LVEDD—left ventricular end diastolic diameter, LVESD—left ventricular end systolic diameter, LAD—left atrial diameter, LVM—left ventricular mass, LVMI—left ventricular mass index, PWTd—posterior wall end diastolic thickness, PWTs—posterior wall end systolic thickness, RWT—relative wall thickness, RVD—right ventricular diameter, TAPSE—tricuspid annulus plane systolic excursion. Data are shown as mean (standard deviation) or median (interquartile range) according to distribution.

**Table 3 medicina-57-00706-t003:** The significant univariate correlations between ARU and studied clinical and echocardiographic parameters.

Variables	Univariate Spearman’s Rho- for Correlation with ARU	*p* Values
Body weight	0.222	0.001
Body mass index	0.207	0.003
Interventricular septal end-systolic thickness	0.170	0.02
Red blood cells distribution width	0.167	0.02
Left ventricular mass	0.153	0.04
NT-proBNP	0.144	0.04
Left ventricular mass index	0.139	0.06
C-reactive protein	0.132	0.06
Prothrombic time	0.136	0.05

NT-proBNP—N-terminal pro-brain natriuretic peptide.

## Data Availability

All materials supporting reported results are available with Kamila Marika Cygulska, cygulskakamila@gmail.com.

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
