# Peer review of "N-Terminal Pro-Brain Natriuretic Peptide and Right Ventricular Diameter Are Related to Aspirin Resistance in Coronary Artery Disease Patients"

_medicina, 2021, doi:10.3390/medicina57070706_

Round 1

Reviewer 1 Report

The authors aimed to reassess the prevalence and predictors of ASAres in contemporary cohort of coronary artery disease (CAD) patients (pts) on stable therapy with ASA, 75 mg o.d. They studied 205 patients with stable CAD treated with daily dose of 75 mg ASA for a minimum of one month. ASAres was detected in 11.7% of pts using point-of-care VerifyNow Aspirin test. Modest correlations were determined between ARU and concentration of NT-proBNP, body weight, body mass index, red blood cell distribution width, left ventricular mass, and septal end-systolic thickness, with trends for left ventricular mass index and prothrombin time. In multivariate regression analysis log(NT-proBNP) was identified as the only independent predictor of ARU.  Authors identified NT-proBNP as the only independent correlate of on-treatment ARU, and a predictor for ASAres, along with right ventricular enlargement and lower hemoglobin concentration in erythrocytes. The article is interesting and relevant. Some things need to be supplemented, corrected and checked.

  1. It seems not correct  results of NT-proBNP are given in tab.1: the result of total patients is 351.2 [136.8-1069.8], but in ASAres group result is 646.3 [309.8-2389.5]. May be an error has crept there. It have to be checked.
  2. The similar situation in Tab. 1 is with creatinine, glucose, glycated hemoglobin, LDL and HDL cholesterol, CRP, WBC, RBC, hemoglobin, hematocrit, mean corpuscular volume, mean corpuscular hemoglobin, RDW, APTT, PT, fibrinogen concentration. The presented data should be checked.
  3. MCHC Asa patients’ result is given not correct: 33.5[33.0-3.9]. It should be corrected.
  4. In the section 2.2 “Platelet function measurement” is not clear, why did 30 min. incubation time was chosen and what were the other incubation conditions.
  5. In the section 2.3 “Clinical data” all blood tests’ methods used should be given and included in the text.
  6. It seems that there are incorrect data in the Tab 2 too: LVEDD, IVSTs, IVSTd, PWTs, PWTd, LVMI, LVM. For example, LVMI is presented in total patients 138.0 [117.0-161.0-max value], but in the ASAres group it is given 147.0 [136.3-167.31- max value in these group is higher than it was given in total patients].
  7. In the Discussion section 4.1 (Heart failure and left ventricular hypertrophy in patients with aspirin resistance) is mentioned, that “PLUTO-CHF trial observed higher platelet activity in patients with CHF treated with ASA“. But there are presented results in the literature, that the highest platelet aggregation is found in NYHA III, when CHF patients were not prescribed any antiaggregant [ Mongirdienė, Aušra;Laukaitienė, Jolanta;Skipskis, Vilius;Kašauskas, Artūras. The Effect of oxidant hypochlorous acid on platelet aggregation and dityrosine concentration in chronic heart failure patients and healthy controls. Medicina. 2019, vol. 55, no. 5 ; https://www.mdpi.com/1010-660X/55/5/198]. It should be cited in the section.
  8. One of the study limitation is that the only one platelet function test was used. It should be mentioned in the “Study limitation section”.

Author Response

Open Review 1

  1. We checked the results of NT-proBNP in tab.1 and they are correct.
  2. We checked other results from table 1 and we made corrections.
  3. MCHC results were corrected.
  4. The incubation time was based on VerifyNow Aspirin Platelet Reactivity Test Instructions. (The information is also included in the article: Nalyaka Sambu , Nick Curzen. Monitoring the effectiveness of antiplatelet therapy: opportunities and limitations. Br J Clin Pharmacol. 2011 Oct;72(4):683-96. doi: 10.1111/j.1365-2125.2011.03955.x.)
  5. Blood tests were performed using the standard method in appropriate test tube.
  6. The results in table 2 were checked and they are correct. The values ​​in square brackets represent the first and third quartiles, not the minimum and the maximum value.
  7. This article was cited in this section.
  8. We mentioned in the “study limitation section” that the only one platelet function test was used.

Reviewer 2 Report

The authors mention in the introduction several clinical correlates with ASA resistance (ref 5-12) but this cannot be found in table 1. The authors should explain the difference between the references and their results.

Some parameters in table 3 (univariate correlation study) show significancies that are not present for the same parameters in tables 1 & 2. The clinical significance should be highlighted.

Author Response

Open review 2

1.      The differences in creatinine results was due to patients selection; in our study there were no significant differences in stage of renal failure [7]. We did not show an association with aspirin resistance and glucose levels probably because few patients had been included in our study with diabetes [10]. We did not confirm a connection with aspirin resistance and age, smoking as in the study by Xian-Feng Liu et al. [6]. We confirm a positive correlation between aspirin resistance and BMI found in the study Ertugrul DT et al. [10].2.      Table 1/2 illustrate our findings regarding variables which discriminated res/sens group, based on dichotomous definition of ASA resistance. Because ARU represents a continuous variable, the significant  correlation do not necessarily reflect the findings from the former analysis. the differences mainly depend on a particular threshold used in the definition of resistance. we agree that NT-proBNP represented significantly in both analyses is probably the most robust finding of our analysis. We were identified several novel correlates of ASA which can contribute to better understanding of the significant phenomenon of aspirin resistance. Our study will allow increasing attention to patients with heart failure, obesity, left ventricular hypertrophy, right ventricular diameter and with unsuccessful aspirin treatment.

Reviewer 3 Report

In the article entitled “N-terminal pro-brain natriuretic peptide and right ventricular diameter are related to aspirin resistance in coronary artery disease patients” by Kamila Cygulska et al., the investigators assessed the prevalence of ASAres in patients with coronary artery disease chronically treated with standard ASA dose, the correlation of on–treatment ARU with clinical and echocardiographic variables and the potential predictors of ASAres. There is soundness to the author's idea, but I have some concerns.

Major

  1. The abstract needs revision:
  • Maximum 300 words
  • It has to be only one paragraph
  • It has to include the following sections: 1) Background and objectives, 2)Materials and methods, 3)Results, 4) Conclusions
  • The TXA2 abbreviation does not need to be entered in the abstract
  • L24 – ”rs” has to be written only ”r”
  1. L53-55 – The phrase ” We were identified several novel 53 correlates of ASAres which can contribute to better understanding of the significant phenomenon of aspirin resistance“ should be introduced better in the personal part, not in the introduction.
  2. In the introduction, there is redundant information regarding the ASAres risk factors reported in the previous publications:
  • BMI: L60 + L64
  • Body wheight : L62 + L64
  1. L72-L73: The text „36.6% females, average age 68.2 (9.7) years, average 72 BMI 27.3 (4.6) kg/m2 ” and Table 1 should be included in the results section and should be removed from the methods section.
  2. The methodology should be completed with:
  • A flow chart that should illustrate the process of enrolling the patients
  • An explanation before L101 which are the two groups of the study. The abbreviation ARAsens from Table 1 needs to be explained here.
  1. Tables 1 and 2:
  • All the parentheses used should be round brackets - „)”
  • Each table should be discussed before inserting it into the text through a comprehensive paragraph. Writing the values into the tables is not enough.

Minor

  1. The title does not have to have a full stop at the end – „ . ”
  2. I do not consider useful the usage of round brackets for the text from L157-159 and L160
  3. There are missing commas throughout the text (e.g. L179)
  4. There are abbreviations used in excess. I recommend shortening words only in situations where those words are repeated at least once. Also, once a word is explained in parentheses with its abbreviation, only the abbreviation will be used later (example of an error in the text for TP – L194, L208, L216)
  5. There are spaces used in excess – L40, 42, 47, 48, 72, 87, 95, 130, 166, 195, 218, 221, 236.
  6. There is text written with another font than the recommended one – L102, 148,161,267, 276
  7. L276 – the grant number should be specified
  8. The bibliographic references on L110, L235 should be correctly inserted
  9. Bibliographic references must respect the font, italic style, or bold style (where required). Please check the instructions for authors on the journal's website.

Author Response

Major

1.      The abstract does not exceed 300 words.

2.      Abstract is only one paragraph.

3.      Abstract includes the following sections: :

1) Background and objectives, 2)Materials and methods, 3)Results, 4) Conclusions

4.      I canceled the TXA2 abbreviation.

5.      We added the phrase L53-55 to the results

6.      We removed the text L72-L73 and Table 1 from the method section and we included in the results.

7.       Fig. 1 Patient inclusion flow diagram

  1. I used 2 types parentheses, because in the round brackets are standard deviations and in the square brackets are interquartile ranges.

Minor

  1. I corrected all suggestions.
  2. I can not specify the grant number, because it was unrestricted research grant from Aflofarm Farmacja sp z o.o.

Round 2

Reviewer 3 Report

There is nothing more I consider it should be added.

It can be accepted in the current form.